# Genes possibly related to symbiosis in early life stages of *Acropora tenuis* inoculated with *Symbiodinium microadriaticum*

Yuki Yoshioka [1,3✉], Yi-Ling Chiu[1], Taiga Uchida [1], Hiroshi Yamashita[2], Go Suzuki[2] & Chuya Shinzato [1✉]

Due to the ecological importance of mutualism between reef-building corals and symbiotic algae (Family Symbiodiniaceae), various transcriptomic studies on coral-algal symbiosis have been performed; however, molecular mechanisms, especially genes essential to initiate and maintain these symbioses remain unknown. We investigated transcriptomic responses of *Acropora tenuis* to inoculation with the native algal symbiont, *Symbiodinium microadriaticum*, during early life stages, and identified possible symbiosis-related genes. Genes involved in immune regulation, protection against oxidative stress, and metabolic interactions between partners are particularly important for symbiosis during *Acropora* early life stages. In addition, molecular phylogenetic analysis revealed that some possible symbiosis-related genes originated by gene duplication in the *Acropora* lineage, suggesting that gene duplication may have been the driving force to establish stable mutualism in *Acropora*, and that symbiotic molecular mechanisms may vary among coral lineages.

[1] Atmosphere and Ocean Research Institute (AORI), The University of Tokyo, Kashiwa, Chiba, Japan. [2] Fisheries Technology Institute, Japan Fisheries Research and Education Agency, Ishigaki, Okinawa, Japan. [3] Present address: Marine Genomics Unit, Okinawa Institute of Science and Technology Graduate University, Onna, Okinawa, Japan. ✉email: y.yoshioka@oist.jp; c.shinzato@aori.u-tokyo.ac.jp

Symbiotic relationships between reef-building corals and dinoflagellates of the family Symbiodiniaceae are essential for coral reefs, the most biologically diverse, shallow-water marine ecosystems[1]. In reef-building corals, symbiotic algae supply the majority of the host's energetic needs via photosynthesis[2–4]. In turn, host corals provide shelter and inorganic nutrients for the algae. However, these symbiotic relationships have begun to collapse in recent decades due to various anthropogenic stresses, including ocean warming associated with climate change[5–7]. These stresses result in coral bleaching, a breakdown of symbiosis between corals and their algal symbionts[8] that ultimately leads to loss of habitat for numerous marine species and that can precipitate the collapse of entire coral reef ecosystems[9].

Most coral species (~71%) acquire algal symbionts from the surrounding environments in each generation[10]. *Acropora* corals, the most common and widespread genus of reef-building corals in the Indo-Pacific[11], tend to harbor dinoflagellates belong to *Symbiodinium* (formerly Clade A) and/or *Durusdinium* (formerly Clade D) in early life stages at Ishigaki Island, Okinawa Prefecture, Japan[12], even though mature colonies generally harbor *Cladocopium* (formerly Clade C)[12–14]. Host-symbiont specificity can also extend to the species level, with *S. microadriaticum* (Smic) predominating among *Symbiodinium* taxa in *Acropora* recruits[15], indicating that Smic is a native symbiont in early life stages of *Acropora* in this region. For this relationship, although high infectiousness of *Symbiodinium* for early life stages of *Acropora* corals is considered one of the reasons why *Symbiodinium* dominates early life stages of *Acropora*[16,17], another possible reason is that *Symbiodinium* tolerates solar radiation and thermal stresses[18–21]. As *Acropora* corals often acquire symbiotic algae from water column during floating larval stages[22], selecting *Symbiodinium* as a symbiotic partner is reasonable. Despite possible advantages of harboring *Symbiodinium*, the reason that mature colonies of *Acropora* harbor *Cladocopium* is probably due to the higher carbon fixation rate of *Cladocopium*, compared with that of *Symbiodinium*[23].

In sea anemone-algal symbioses, after recognition of symbiotic partners, downstream cellular signaling pathways, such as the innate immune system, are modulated to initiate symbiosis[24], and metabolic interactions between the partners occur[25]. In studies using sea anemones, their native algal symbionts have been generally used in inoculation experiments, e.g., for experiments with *Exaiptasia pallida*, *Breviolum minutum* strain SSB01 isolated from *E. pallida*[26–28]. For coral-algal symbioses, several studies have examined coral transcriptomic responses to dinoflagellates (reviewed in Meyer & Weil[29]), but cellular mechanisms are still unclear because those studies used dinoflagellate strains that were not isolated from the corals in question. The main reason is the difficulty of establishing culture strains of dinoflagellates isolated from corals. Two studies used Symbiodiniaceae strains isolated from corals for inoculation experiments. One study inoculated *A. tenuis* polyps with an algal strain (*D. trenchii*) isolated from the coral, *Montastraea*, which is phylogenetically distinct from *Acropora*[30], and another inoculated *A. tenuis* larvae with *Cladocopium goreaui*, isolated from fragments of an *A. tenuis* colony[31]. However, algal strains used in these two studies are not native symbionts in early life stages of *A. tenuis*. Therefore, it is unclear whether the reported molecular responses could occur in natural symbiotic relationships in corals. In addition, no comparison of differentially expressed gene (DEG) repertoires has been performed because earlier studies used transcriptome assemblies that were created independently[30,31]. For *A. tenuis*, we are now able to access a whole-genome assembly and high-quality gene models[32], allowing direct comparisons of DEG repertoires across studies.

We recently developed an *Acropora* larval system as a model to study symbiont selection and recognition by host corals[33]. Using this system, we previously revealed larval transcriptomic responses of *A. tenuis* to Smic (strain AJIS2-C2 isolated from an *Acropora* recruit) from initial contact to symbiosis establishment and identified genes specifically involved in symbiosis with Smic[34,35]. However it is unclear whether the larval symbiosis-related genes continue to contribute to symbiosis at the polyp stage, the next developmental stage of planula larvae. In addition, it is challenging to distinguish between genes associated with symbiosis and genes involved in development in corals that acquire algal symbionts horizontally in early life stages. It is expected that key symbiosis-related genes will be continuously expressed or suppressed when corals acquire algal symbionts. Therefore, it is crucial to validate the candidate genes involved in symbiosis with multiple developmental stages. In the present study, we performed inoculation experiments with Smic using *A. tenuis* primary polyps and analyzed transcriptomes of *A. tenuis* primary polyps inoculated with Smic to reveal transcriptomic responses of primary polyps to Smic inoculation. Then we compared transcriptomic responses of *A. tenuis* planula larvae and primary polyps inoculated with Smic to identify genes associated with symbiosis in *A. tenuis* early life stages.

## Results

**Genes responding to Smic inoculation in *A. tenuis* primary polyps.** We performed inoculation experiments with Smic using *A. tenuis* primary polyps that lacked algal cells (Fig. 1a). Successful infection with Smic was confirmed by fluorescence microscopy 10- and 20-days post-inoculation (dpi) in *A. tenuis* primary polyps. Uptake efficiency of Smic in *A. tenuis* primary polyps was 100% at 10 and 20 dpi (Supplementary Table 1). Cell densities (average ± SE cells per polyp) of Smic in *A. tenuis* primary polyps were 1138.9 ± 249 and 4241 ± 872 cells/polyp at 10 and 20 dpi, respectively (Supplementary Table 1), indicating that the number of Smic increased in primary polyps between 10 and 20 dpi when the Smic supply was stopped, as with planula larvae inoculated with Smic[33]. Not surprisingly, algae-infected primary polyps were not observed in the control without addition of Smic. Thus, there was no unexpected Smic contamination during the experimental period.

In order to identify transcriptomic changes of *A. tenuis* primary polyps in response to inoculation with Smic, we performed 3′ mRNA sequencing of *A. tenuis* primary polyps inoculated with and without Smic exposure. An average of ten million RNA-seq reads per sample were retained after quality trimming, ~60% of which were mapped to *A. tenuis* gene models ($n = 3$ for 10 dpi, $n = 2$ for 20 dpi; Supplementary Table 2). Then we compared gene expression levels between Smic exposed and unexposed groups and identified 9 (8 upregulated and 1 downregulated) and 58 DEGs (34 upregulated and 24 downregulated) at 10 and 20 dpi, respectively (Fig. 1b; Supplementary Data 1). Among these, 75% (6/8), 100% (1/1), 68% (23/34), and 33% (8/24) had a homology with genes in the Swiss-Prot database (Supplementary Data 1). Only five genes (aten_s0035.g9, aten_s0035.g10, aten_s0042.g26, aten_s0050.g68, and aten_s0342.g22) were differentially expressed at both 10 and 20 dpi, all of which were upregulated at both sampling timepoints (Fig. 1b; Supplementary Data 1).

**Comparison of DEG repertoires between primary polyps and planula larvae inoculated with Smic.** In order to identify genes associated with symbiosis shared by multiple developmental stages of *A. tenuis*, we compared repertoires of genes whose expression levels were significantly (false discovery rate (FDR) < 0.05) altered by Smic inoculation in primary polyps and planula

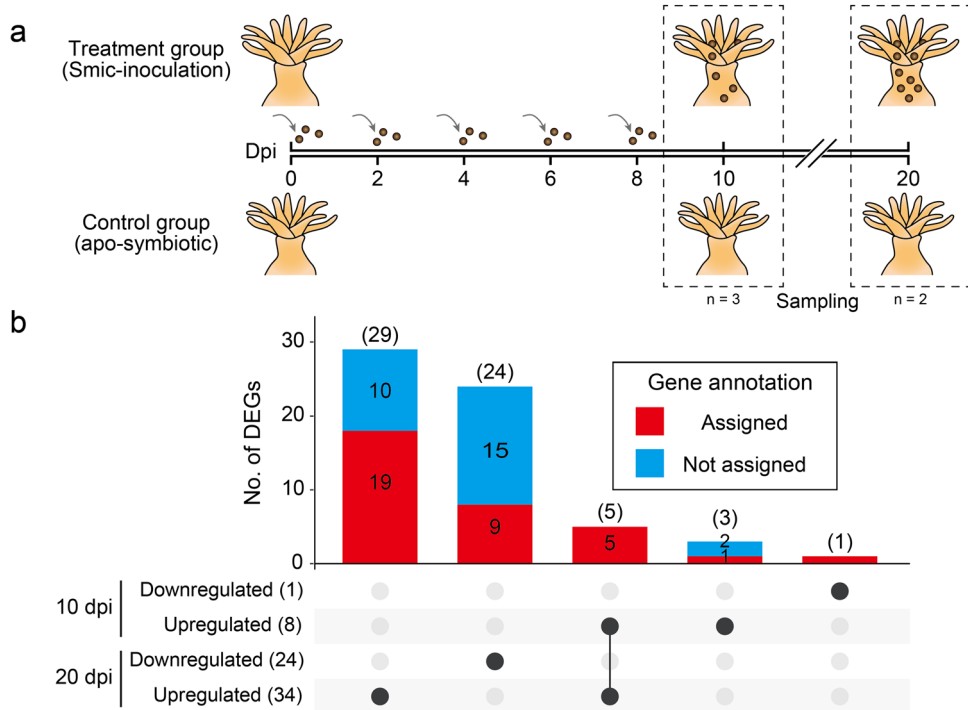

**Fig. 1 Experimental design and UpSet plot showing the number of differentially expressed genes of *Acropora tenuis* primary polyps inoculated with Smic. a** A schematic illustration of the experimental design. Primary polyps were maintained at a concentration of ~5 individuals/well of a 24-well multi-plate. 200,000 algal cells of Smic were added to the treatment group. Two days after addition of algae, artificial seawater (ASW) was replaced, and 200,000 algal cells of Smic were reintroduced. The process of ASW exchange and algal addition was repeated every two days until 8 days post-inoculation (dpi). Subsequently, only ASW exchange was conducted until 20 dpi. RNA extraction was conducted for each well containing ~5 primary polyps at 10 and 20 dpi. One well containing ~5 individuals was considered one replicate. See Method for details. **b** Total numbers of up- and downregulated DEGs in each dpi are shown on the left. Each row contains elements represented by black circles. The line connecting elements indicates DEGs shared between samples. Vertical bars with blue and/or red indicate results of DEG annotation (BLAST searches against the Swiss-Prot database) belonging to each element. Gene ID, Swiss-Prot annotation, and Pfam domain of DEGs are provided in Supplementary Data 1.

larvae. For gene repertoires in planula larvae, we used RNA sequencing data of *A. tenuis* planula larvae inoculated with Smic at 4, 8, and 12 dpi and their corresponding apo-symbiotic larvae[34]. We found that expression levels of 3315 genes were significantly altered (FDR < 0.05) by Smic inoculation in planula larvae (Supplementary Data 1). When we compared DEG repertoires between planula larvae and primary polyps inoculated with Smic, 17 DEGs were common between them (Fig. 2a). Furthermore, to test whether expression levels of these DEGs are correlated with numbers of algal symbionts, we performed correlation analysis between numbers of algal cells in corals (Fig. 2b; Supplementary Table 1) and their expression levels (Supplementary Data 1). Expression levels of 15 DEGs showed a positive correlation with numbers of algae in corals ($r > 0.6$), while the remaining two genes showed no correlation (Fig. 2c; Supplementary Fig. 1). Transcriptomic similarity of alga-containing cells in planula larvae and primary polyps has been suggested on the basis of whole-organism single-cell analysis of the coral, *Stylophora pistillata*[36], suggesting that these 15 genes may have similar functions in planula larvae and primary polyps inoculated with Smic. We refer to these 15 genes as possible symbiosis-related genes in *A. tenuis* early life stages in this context. Of these, gene functions previously thought to be involved in coral-algal symbiosis, such as sugar transport (*SLC2A8*)[34,36,37], lipid transport (*NPC2*)[34,36,38], sulfate transport (*SLC26A2*)[30,34], protection against oxidative stress (*Pxd, Chac1, PSAP*-like, and *MMP*-like)[34,36,39,40], immune system (*Grn*-like)[41,42], and Notch-related signaling pathway (*AnNLL*)[34] were included (Fig. 2c), implying

that these 15 genes serve crucial functions during symbiosis in corals.

On the other hand, 3298 and 45 DEGs were restricted to planula larvae and primary polyps inoculated with Smic, respectively (Fig. 2a). These DEGs may contribute to each developmental stage with algal symbionts. For example, 2519 genes (1858 + 285 + 376) of 3298 DEGs (~76%) specifically detected in planula larvae inoculated with Smic were specific to each sampling point, 4, 8, or 12 dpi (Supplementary Fig. 2a). In addition, expression levels of 6135 genes differed (FDR < 0.05) depending on age (days) in apo-symbiotic planula larvae, 1605 of which were differentially expressed in planula larvae inoculated with Smic (Supplementary Fig. 2b). In contrast, expression levels of 63 genes differed depending on age in apo-symbiotic primary polyps, 3 of which were differentially expressed in primary polyps inoculated with Smic (Supplementary Fig. 2c; d). Major differences in appearance between coral larvae and primary polyps are the presence of skeletons and tentacles in primary polyps. Genes such as a possible skeletal organic matrix protein (*USOMP*: aten_s0047.g56), genes for proteins that transport ions or amino acids required for skeleton formation (*KcnJ4*: aten_s0110.g47and *SLC32A1*: aten_s0205.g30), and a possible tentacle formation-related gene in *Hydra*[43] (*Wnt8b*-like: aten_s0336.g16) are predicted to be involved in growth accelerated by symbiosis in primary polyps (Supplementary Data 1). On the other hand, a gene previously reported to be related to symbiosis was also included. *SCARB2*-like gene (aten_s0190.g17) with a scavenger receptor domain CD36, one

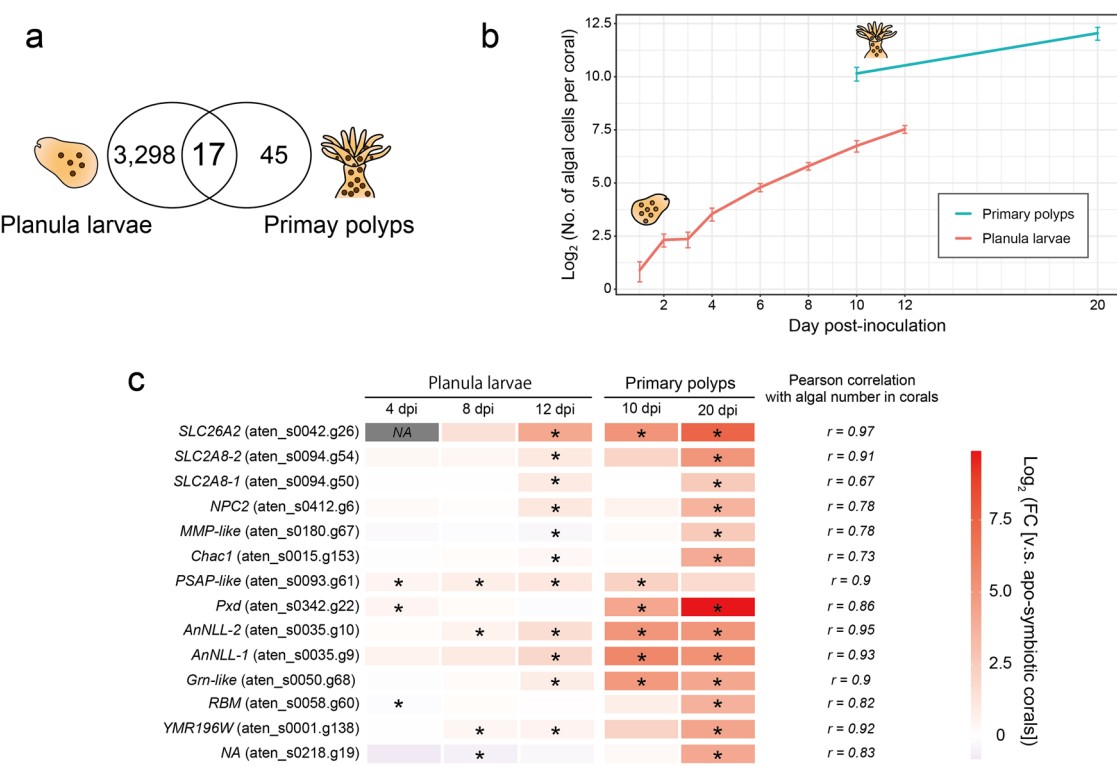

**Fig. 2 Identification of possible symbiosis-related genes during symbiosis in early *Acropora tenuis* life stages. a** A Venn-diagram of the number of DEGs in planula larvae and primary polyps inoculated with Smic. **b** Numbers of acquired Smic cells per coral. Lines indicate average ± SE of acquired Smic cells in primary polyps and planula larvae. Raw data are shown in Supplementary Table 1. **c** Heatmap showing log₂-transformed fold change of 15 possible symbiosis-related DEGs whose expression levels correlated with numbers of algae in corals. Expression levels were compared to corresponding controls (apo-symbiotic larvae or primary polyps). Asterisks indicate significant differences of gene expression levels (FDR < 0.05) compared with controls. NA indicates that calculations were not possible due to very low expression in control samples.

of the pattern-recognition receptors, is thought to be important in communication with algal symbionts[44,45] and immune modulation in sea anemone-algal symbiosis[46]. Recent studies revealed that a gene with a CD36 domain was also strongly expressed in alga-containing cells in the coral, *S. pistillata*[36] and an octocoral *Xenia*[47], suggesting that *SCARB2*-like gene is also important in *A. tenuis* primary polyps during symbiosis.

**Detailed analyses of possible symbiosis-related genes in early life stages of *Acropora tenuis*.** In order to deduce protein functions of the 15 possible symbiosis-related genes, we first analyzed whether they have evolutionary conserved protein domains. At least one evolutionarily conserved protein domain was detected from 12 of the 15 genes (Fig. 3). We next analyzed whether their potential homologs are present in model organisms (human, fruit flies, or yeast). Among the 12 genes, 10 had homology to human genes (aten_s0015.g153; aten_s0042.g26; aten_s0050.g68; aten_s0058.g60; aten_s0093.g61; aten_s0094.g50; aten_s0094.g54; aten_s0180.g67; aten_s0342.g22; aten_s0412.g6) and domain compositions of 7 genes were identical to those of human homologs (Supplementary Fig. 3). Although no homolog of aten_s0001.g138 was identified in the human genome, its potential homolog was identified in yeasts, and the gene contained evolutionarily conserved protein domains that are the same as the homolog from yeasts (Supplementary Fig. 4). These results suggest that functions of 8 proteins are similar to human or yeast homologs, functions of which are known. Although 3 genes (aten_s0093.g61, aten_s0050.g68, and aten_s0180.g67) had evolutionarily conserved protein domains in the sequences, they

lacked some protein domains compared with human genes (Supplementary Fig. 3). In addition, the missing protein domains were not identified, even under relaxing protein domain searches (PHMMER, E-value cutoff: 1), suggesting that mutations may have accumulated in these protein sequences during evolution. Indeed, one of the 3 genes, aten_s0093.g61, was suggested as a rapidly evolving gene in *Acropora* lineages[32].

Not surprisingly, potential homologs of the 15 possible symbiosis-related genes were detected in 11 other cnidarian genomes, but their copy numbers differed among species or lineages (Fig. 3). For aten_s0218.g19 and aten_s0219.g19, no evolutionarily conserved protein domains were identified among protein sequences, even under relaxed searches (PHMMER, E-value cutoff: 1), although their homologs were detected in several coral lineages (Fig. 3), suggesting that they are lineage-specific genes and may contain novel protein domains that have not yet been identified.

In order to illuminate the evolutionary histories of the 15 possible symbiosis-related genes, we performed molecular phylogenetic analysis with either their potential homologs in humans, fruit flies, or yeasts. This analysis revealed that aten_s0001.g138 (*YMR196W*), aten_s0015.g153 (*Chac1*), aten_s0042.g26 (*SLC26A6*), aten_s0058.g60 (*RBM*), aten_s0094.g50 (*SLC2A8*), aten_s0094.g54 (*SLC2A8*), aten_s0342.g22 (*Pxd*), and aten_s0412.g6 (*NPC2*) are clustered with their potential homologs in humans, fruit flies, or yeasts (Supplementary Figs. 4–10), indicating that they are possible orthologs. On the other hand, 3 genes (aten_s0050.g68; aten_s0093.g61; aten_s0180.g67) were not clustered with potential homologs in humans or fruit flies (Supplementary Figs. 11–13), but since they had protein domains similar to those of humans or fruit

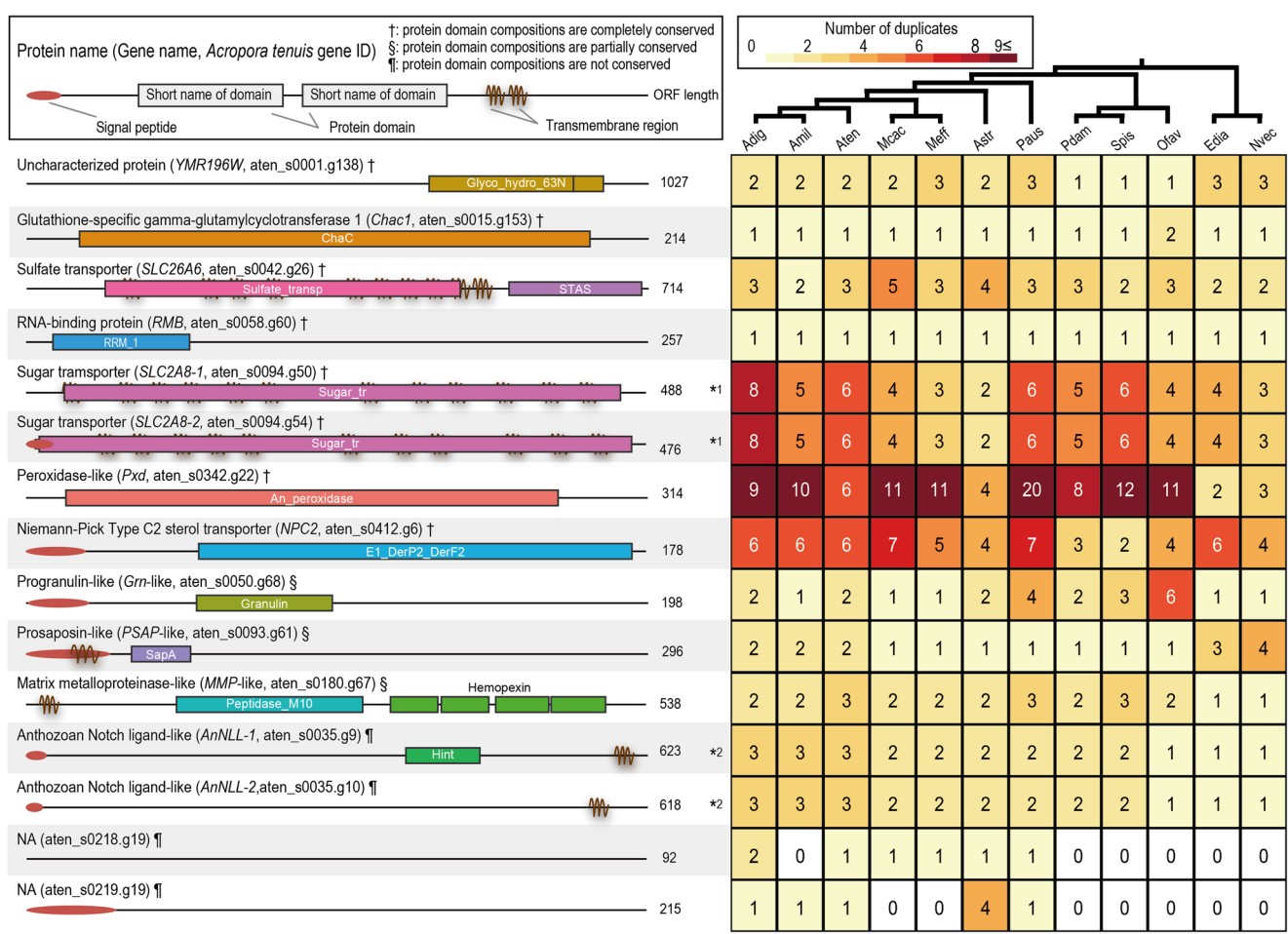

**Fig. 3 Molecular features of possible symbiosis-related genes during symbiosis in early *Acropora tenuis* life stages.** Possible protein name, gene name, and gene ID are shown at the left of each sequence (Left). Short names of evolutionary conserved protein domains (Pfams) are shown in or around each domain. Amino acid sequence length is shown at the right of each sequence. Symbols (†, §, and ¶) after parentheses indicate that protein domain compositions are completely conserved, partially conserved, or not conserved, compared with homologs of humans or yeasts. Numbers of duplicates are shown in the heatmap (Right). Tree topology shows phylogenetic relationships among species inferred by OrthoFinder. *1 and *2 indicate that they belong to the same gene family. Abbreviation: Aten, *A. tenuis*; Adig, *A. digitifera*; Amil, *A. millepora*;, Mcac, *M. cactus*; Meff, *M. efflorescens*; Astr, *Astreopora myriophthalma*; Paus, *Porites australiensis*; Spis *S. pistillata*; Ofav, *O. faveolata*; Pdam, *Pocillopora damicornis*; Edia, *Exaiptasia diaphana*; Nvec, *Nematostella vectensis*.

flies, we refer to them as *Grn*-like (aten_s0050.g68), *Psap*-like (aten_s0093.g61), and *MMP*-like (aten_s0180.g67) in this context. Our previous study revealed that aten_s0035.g9 and g10 are anthozoan Notch ligand-like genes (*AnNLL*)[34] (Supplementary Fig. 14). This analysis also demonstrated that *SLC2A8-1*, *SLC2A8-2*, *Pxd*, *Psap*-like, *MMP*-like, *AnNLL-1*, and *AnNLL-2* have been expanded by tandem duplication in the *A. tenuis* genome, and counterparts were also found in two other *Acropora* genomes (Supplementary Figs. 8b; 9b; 12b; 13b; 14b), indicating that the tandem duplication occurred in the common ancestor of *Acropora*.

Furthermore, to test whether the 15 possible symbiosis-related genes are also associated with symbiosis in adult stage of *A. tenuis*, we compared expression levels of the 15 genes in several developmental stages (ranging from eggs to polyps) with no symbiotic algae and a mature colony containing symbiotic algae. This comparison revealed that expression levels of nine genes (*YMR196W*; *AnNLL*-1; *AnNLL*-2; *SLC26A6*; *Grn*-like; *Psap*-like; *SLC26A6-1*; *SLC26A6-2*; *NPC2*) are more highly expressed (Z score > 1.96) in *A. tenuis* mature colonies than in developmental stages with no symbiotic algae (Supplementary Fig. 15), suggesting that these genes are important throughout the entire life cycle of *A. tenuis* harboring symbiotic algae.

**Discussion**

In this study, we identified 9 and 58 DEGs in primary polyps inoculated with Smic (Fig. 1b). In contrast to the low numbers of DEGs, a larger number of DEGs were identified in planula larvae inoculated with Smic (Supplementary Data 1). A possible explanation for this is the presence of large number of genes whose expression levels are controlled by developmental stage or age (days), rather than by algal infection (Supplementary Fig. 3). In other words, a substantial portion of DEGs identified in planula larvae may have arisen as an indirect effect of algal inoculation. In contrast, DEGs identified in primary polyps may represent genes directly related to algal inoculation. On the other hand, Yuyama et al.[30] reported differential expression of 7667 genes in *A. tenuis* primary polyps inoculated with *D. trenchii*, a much higher number of DEGs than in this study. Although it is difficult to directly compare studies using different algal symbiont types, it is important to note that algal infection affects circadian rhythms of coral hosts regardless of algal species[34]. In addition, algal culture media contain different types of nutrients and prolonged exposure to residual algal culture media may cause significant differences in polyp development. Indeed, asexual reproduction by budding was observed in primary polyps inoculated with *D. trenchii*, whereas no budding occurred in apo-symbiotic polyps[30].

Note that algal culture media was replaced with ASW during inoculation experiments in this study (see Methods). Comparing gene expression between samples in different states poses a challenge in determining whether observed changes in gene expression are truly associated with symbiosis. Therefore, it is crucial to validate results using multiple developmental stages to identify genes involved in coral symbiosis, and the 15 genes that commonly upregulated in planula larvae and primary polyps inoculated with Smic are likely associated with symbiosis in early life stages of *A. tenuis*.

When *A. tenuis* planula larvae were inoculated with Smic, they downregulated gene expression involved in metabolism and upregulated transporter genes[34]. On the other hand, in this study, when *A. tenuis* primary polyps were inoculated with Smic, they also upregulated transporter genes, but did not downregulate gene expression involved in metabolism. Harii et al.[48] reported that survival rate and lipid levels of planula larvae with symbiotic algae increased under insolation, confirming that planula larvae assimilate photosynthetic products from algal symbionts. The planula larval stage is the only dispersal stage for corals. For many marine organisms, ocean currents influence larval dispersal[49], suggesting that long-term floating on the ocean surface enables long-distance dispersal. Therefore, the strategy of slowing metabolic rates and assimilating photosynthetic products from algal symbionts may promote long-term dispersal of coral planula larvae, which is important for linking and maintaining populations. Therefore, it is likely that utilization of photosynthetic products produced by symbiotic algae without reducing metabolic rates would allow for faster growth and increased fitness of primary polyps.

It is believed that more than 90% of host coral energy needs are provided by nutrients such as sugars and amino acids produced by algal symbionts photosynthetically; hence, host genes involved in nutrient transport are thought to be upregulated during symbiosis[41]. These metabolic interactions are one of the central physiological processes of symbiosis[25]. Two possible symbiosis-related genes (aten_s0094.g50 and g54) were clustered with human *SLC2A6* and *SLC2A8*, which transport glucose[50] and aten_s0412.g6 clustered with human *NPC2*, which transports cholesterol[51] (Supplementary Figs. 8; 10). Glucose is thought to be the major metabolite transferred from symbiotic algae to host corals[52,53]. Upregulation of sugar transporters under illuminated conditions has been reported in *Acropora* colonies[37] and in alga-containing cells of the coral *S. pistillata*[36], suggesting that it is an important energy source for corals (Fig. 4). Lipids, including sterols, are also thought to be translocated from symbiotic algae to host corals[54–56]. It has been suggested that the majority of marine invertebrates, including corals, are unable to synthesize sterols because they may have lost a necessary enzyme, essential for cell membrane synthesis[57,58], indicating that corals must acquire sterols from the diet and/or symbionts. Cnidarians possess at least two types of sterol transporters, *NPC 1* and *NPC2*, and NPC2 protein is more highly expressed in symbiotic than in non-symbiotic sea anemones[38,45,59]. Furthermore, NPC2 protein was also localized in symbiosomes, host organelles that harbor symbiotic algae, in sea anemones[38], suggesting that *NPC2* may serve as a sterol transporter in *Acropora* during symbiosis and may be localized in symbiosomes in alga-containing coral cells (Fig. 4).

In addition to transporters that facilitate translocation of nutrients from algal symbionts to host corals, transporters that enable the transfer of nutrients in the opposite direction (from host corals to algal symbionts) would be necessary to maintain coral-algal symbiosis. One possible symbiosis-related gene (aten_s0042.g26) was clustered with human *SLC26A1* and *SLC26A2*, which transport sulfate[60,61] (Supplementary Fig. 6).

Earlier studies also reported upregulation of sulfate transporters in corals during symbiosis[30,62]. Although complete sulfur metabolism in corals remains unclear, it has been hypothesized that sulfate ions are transferred from host corals to symbiotic algae, and sulfur-containing amino acids (cysteine and methionine) are synthesized by symbiotic algae and returned to their hosts[63]. Cystathionine β-synthase, an enzyme essential for cysteine biosynthesis, has not been detected in *Acropora* genomes[32,64]. Although a recent study using a yeast complementation assay suggested that cysteine biosynthesis is possible in *A. loripes* with an alternative pathway[65], cysteine biosynthesis was not confirmed in *A. solitaryensis* using sulfur isotope labeling in culture[66], suggesting that *Acropora* corals may still rely upon algal symbionts and/or diet for cysteine. In addition, cysteine is required for synthesis of glutathione, an important antioxidant[67]. Therefore, the sulfate transporter may be important for sulfur metabolism in early *Acropora* life stages (Fig. 4).

In this study, 4 of 15 possible symbiosis-related genes (*Chac1*; *Psap*-like; *MMP*-like; *Pxd*) possibly involved in defense against oxidative stress were detected (Fig. 3). Coenzyme Q and glutathione are two of the antioxidants in corals[39,68]. Prosaposin, encoded by *Psap*, may regulate coenzyme Q10 in humans[69], and glutathione-specific gamma-glutamylcyclotransferase, encoded by *Chac1*, specifically acts on glutathione in humans[70]. The possibility of metal accumulation by glutathione, which binds metals has been suggested in sea anemones[71,72]. Peroxidase, encoded by *Pxd*, is one of the antioxidant enzymes that converts reactive oxygen species (ROS) back to oxygen and water and is thought to be important in coral-algal symbioses[39]. Oxidative stress likely induces symbiosis dysfunction, leading to coral bleaching. It is caused by ROS produced by host mitochondrial membranes and/or the photosynthetic apparatus in algal symbionts[73]. To prevent cellular damage caused by ROS, it has been suggested that corals express antioxidants and/or antioxidant enzymes[68,73]. Although algal density in host corals is controlled within a certain range ($0.5–0.75 \times 10^6$ algae per mg host protein) in adult corals[74], due to algal proliferation, the number of algae in host corals continues to increase until it reaches a certain number in early life stages (Fig. 2b; Supplementary Table 1) or the accumulation of engulfed algae from the environment, indicating that the total amount of ROS also increases. Therefore, corals need to express antioxidants when they acquire algal symbionts in early life stages to prevent cellular damage (Fig. 4).

Innate immunity system is important in cnidarian symbioses[24,41]. After engulfment of appropriate algal symbionts by host coral cells, hosts suppress immune responses in order to initiate symbiosis[26,35,41]. In contrast, many immune-related genes are upregulated during symbiosis dysfunction[24]. These responses indicate that host cnidarians are likely associated with down-regulation of immune-related pathways at onset and during maintenance of symbiosis[24]. In sea anemone-algal symbiosis, immune stimulation inhibits infection by native symbionts[26]. However, once symbionts are stably integrated in symbiosomes, stability of symbiosis is not compromised by artificial immune stimulation[26]. Experimental evidence suggests that a specific protein, lysosomal-associated membrane protein 1, decorating the symbiosome membrane, contributes to this stability[26]. It is likely that proteins that localize on and interact with the symbiosome membrane of the symbiosome may managing the immune system. Progranulin, encoded by *Grn*, is a multifunctional protein that modulates immune responses in mammals, acting as a ligand to block TNF-induced immune responses[75]. In this study, a *Grn*-like gene was identified as a possible symbiosis-related gene (Fig. 3). Previous studies have reported higher expression levels of a *Grn*-like gene in alga-containing cells of *S. pistillata*[36] and *Xenia*[47]. Another possible symbiosis-related gene identified in

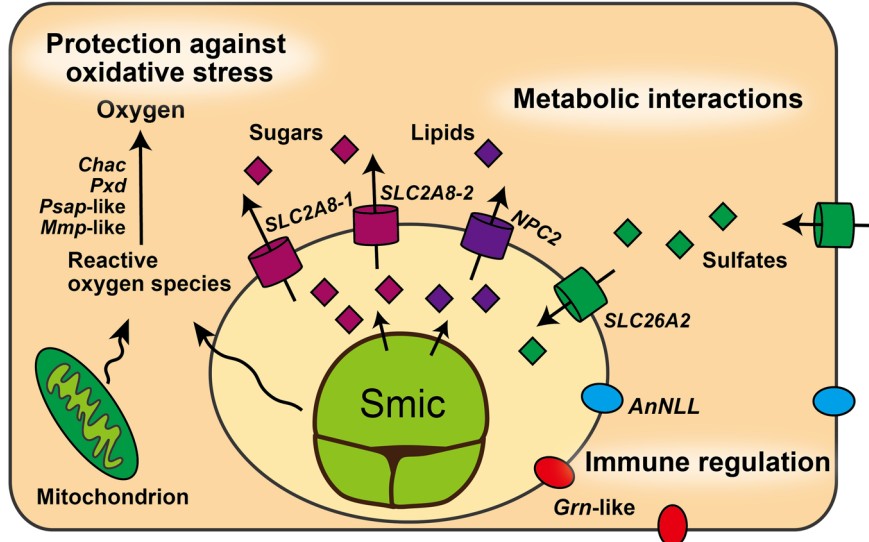

**Fig. 4 Schematic summary of hypothetical functions of symbiosis-related genes in *A. tenuis* early life stages during symbiosis with the native symbiont, *S. microadriaticum*.** The symbiotic algae Smic (*S. microadriaticum*) is surrounded by the symbiosome, the organelle in which a symbiont resides. Antioxidant response regulatory genes (*Chac*, *Pxd*, *Psap*-like, and *Mmp*-like) are expected to protect against oxidative stress caused by reactive oxygen species produced in coral mitochondria and/or plastids of algal symbionts. Transporter genes (*SLC2A8*, *SLC26A2*, and *NPC2*) are likely involved in metabolic interactions of sugars, lipids, and sulfates between symbiotic partners, Ligand-like genes (*AnNLL* and *Grn*-like) are predicted to be involved in immune regulation for maintenance of coral-algal symbioses.

this study, *AnNLL*, has been predicted to function as a ligand for the Notch signaling pathway[34]. These two ligand-like genes may interact with a receptor on symbiosome membranes and/or alga-containing cells to promote persistence of symbioses (Fig. 4).

Gene duplication is a common phenomenon and contributes to genetic novelty[76]. In artificial evolution experiments, gene duplication of transporters enhanced affinity and uptake velocities[77]. Importance of duplicated genes has been suggested in symbiotic insects, e.g., acquisition of specific gene functions in bacteriocytes (bacteria-containing cells)[78–80]. In this study, we showed that some possible symbiosis-related genes originated by gene duplication in coral genomes (Fig. 3). The common ancestor of reef-building corals appeared a hundred million years ago, and between coral-algal symbiosis is thought have occurred before that[81]. Interestingly, some possible symbiosis-related genes were duplicated after divergence of extant shallow-water reef-building corals into two major clades (Robusta/Complexa)[82], suggesting that not only genes duplicated in the common ancestor of reef-building corals, but also genes duplicated after speciation also involve symbiosis. That finding also suggests that molecular mechanisms of symbiosis in corals are diverse and may be related to specificity between host corals and symbiotic algae, e.g., symbiotic dinoflagellates (species or genetic types) tend to be unique to each coral familial or generic lineage[83]. Similarly, a recent study using mathematical modeling also suggested lineage-specific variation in coral-algal symbiosis[84]. Therefore, gene duplication may have been a driving force to establish mutualism in each coral-lineage.

## Methods

**Preparation of *A. tenuis* primary polyps.** Colonies of *A. tenuis* were collected in Sekisei Lagoon, Okinawa, Japan in May 2022, and were maintained in aquaria at the Yaeyama Station, Fisheries Technology Institute, until spawning. Permits for coral collection were kindly provided by the Okinawa Prefectural Government (Permits 3–67 [1]). After fertilization, we washed embryos with 0.2-μm filtered seawater (FSW) to remove contaminants. Embryos were maintained in plastic bottles at 27 °C. FSW was

changed once a day for 5 days. After 5 days, planula larvae were transferred to AORI, The University of Tokyo, and metamorphosis was induced by exposure of larvae to 100 mM Hym 248 (development regulating peptide) in 24-well multi-plates, as in Iwao et al.[85]. Metamorphosed larvae were maintained at a concentration of ~5 individuals/mL of artificial sea water (ASW) prepared with Viesalt (Marine Tech) at 25 °C until they became polyps.

**Algal inoculation experiments using *A. tenuis* primary polyps.** Two days after metamorphosis, we divided primary polyps into two treatment groups, with at least two wells per treatment. One well containing ~5 individuals was considered one replicate in this study (Fig. 1a). Smic AJIS2-C2 strain was used in this study. To remove algal culture medium, Smic was centrifuged at 3000 rpm for 5 min. The supernatant was discarded, and the pellet was resuspended with 50 mL of ASW. This wash was repeated once more. Algal concentration was adjusted to $1.0 \times 10^5$ cells/mL, and 2 mL of algae-containing ASW (200,000 algal cells) were added to the first group of primary polyps (treatment group). For the remaining group, 2 mL of ASW were added, which was used as a control group (apo-symbiotic). All bottles were kept at 25 °C under 12-h light (~10 μmol photons m$^{-2}$ s$^{-1}$) and 12-h dark cycle. Two days after addition of algae, ASW was exchanged, and 200,000 algal cells of Smic were re-added. ASW exchange and addition of algae were repeated every two days until 8 dpi. After 8 dpi, only ASW exchange was repeated until 20 dpi. At 10 and 20 dpi, five randomly selected polyps from each well of a 24-well multi-plate were selected, and coral cells were dissociated with 0.5 % Trypsin-EDTA (Cat-No. 15400054, Thermo Fisher Scientific). Subsequently, the number of algae in polyps (uptake efficiency) was counted with a hemocytometer using fluorescence microscopy.

**RNA extraction, sequencing of primary polyps inoculated with Smic.** At 10 and 20 dpi, ~5 primary polyps from each well of 24-well dishes were collected ($n = 3$ for 10 dpi and $n = 2$ for 20 dpi) and homogenized with zirconia beads (TOMY ZB-20) in TRIzol reagent (Thermo Fisher Scientific) using a bead beater (TOMY

Micro Smash MS-100) at 3000 rpm for 10 s. Total RNA was extracted from polyps using TRIzol reagent according to the manufacturer's protocol and were purified with an RNeasy Mini Kit (Qiagen). A Collibri 3' mRNA Library Prep Kit for Illumina (Thermo Fisher Scientific) was used for sequencing library preparation. Sequencing adaptors were attached by PCR amplification with 16 cycles of annealing according to the manufacture's protocol. Each library was sequenced on a NovaSeq 6000 (Illumina) with 50-bp, single-end reads.

**Transcriptomic analyses.** Low-quality reads (quality score <20 and length <20 bp) and Illumina sequence adaptors were trimmed with CUTADAPT v1.16[86]. Then, cleaned reads were mapped to *A. tenuis* gene models[32], downloaded from the genome browser of the OIST Marine Genomics Unit (https://marinegenomics.oist.jp), using BWA v0.7.17[87] with default settings. Transcript abundances in each sample were quantified using SALMON v1.5.2[88] with default settings. Mapping counts were normalized by the trimmed mean of M values (TMM) method, and then converted to counts per million (CPM) using EdgeR v3.32.1[89] in R v4.0.3[90]. Gene expression levels (CPM) in treatment groups were compared with control samples (apo-symbiotic polyps) to identify differentially expressed genes (DEGs). Obtained $p$ values were adjusted using the Benjamini-Hochberg method in EdgeR. When the gene expression level was significantly different (FDR < 0.05) from control samples, genes were considered DEGs. *A. tenuis* gene models were annotated using homology searches with BLAST v2.2.28 (BLASTP, E-value cutoff: 1e-5)[91] against the Swiss-Prot database and protein domain searches with hidden Markov models (InterProScan v5.27-66.0, E-value cutoff: 1e-3)[92] or with HMMER web server (E-value cutoff: 1)[93] against the Pfam database. Putative transposable elements in gene models were identified with Pfam domain ("Transposase", "Integrase", and "Reverse transcriptase") and were excluded from subsequent analyses.

**Comparison of DEG repertoires between *A. tenuis* larvae and polyps inoculated with Smic.** RNA-seq data of *A. tenuis* planula larvae inoculated with and without Smic exposure (apo-symbiotic larvae) at 4, 8, and 12 dpi[34] were used. Data processing was performed as in Yoshioka et al.[34]. Briefly, RNA-seq reads were mapped to *A. tenuis* gene models after trimming of low-quality reads and Illumina sequence adaptors. Gene expression levels of Smic-inoculated samples were compared to their corresponding apo-symbiotic samples to identify DEGs. We defined DEGs as significantly different (FDR < 0.05) compared with apo-symbiotic larvae in this study. These DEG repertoires were compared with those identified in *A. tenuis* primary polyps inoculated with Smic. A list of DEGs is provided in Supplementary Data 1.

**Correlation analysis between gene expression levels and algal densities in corals.** Relative gene expression levels of symbiotic planula larvae and primary polyps compared to their corresponding apo-symbiotic larvae and polyps were calculated using EdgeR. Correlation between $\log_2$ (relative gene expression levels of corals inoculated with Smic) and $\log_2$ (algal cell numbers in corals) were examined using the function "cor.test" in R[90]. Genes with correlation coefficient >0.6 were considered positively correlated with algal number in corals. Relative gene expression levels of symbiotic planula larvae and primary polyps are provided in Supplementary Data 1. Numbers of algal cells in corals are provided in Supplementary Table 1.

**Identification of age-dependent DEGs in primary polyps and planula larvae.** Gene expression levels between apo-symbiotic primary polyps at 10 and 20 dpi (this study) and between apo-symbiotic planula larvae at 4, 8 and 12 dpi (Yoshioka et al.[34]) were compared pairwise using EdgeR, respectively. When gene expression levels differed significantly (FDR < 0.05) between apo-symbiotic larvae/polyps at different ages, those were considered genes that differed by age.

**Quantification of gene expression levels at various developmental stages of *A. tenuis*.** Publicly available RNA-seq data of *A. tenuis* in various samples, including eggs, blastulae, gastrulae, early planula larvae, planula larvae, polyps, and an adult colony (Accession number: DRA011820[94]) were used. Data processing was performed as in Yoshioka et al.[94]. Briefly, low-quality reads and sequencing adaptors were trimmed, and cleaned reads were mapped to *A. tenuis* gene models. Gene expression levels (transcripts per million, TPM) were estimated with SALMON and were transformed into Z scores. The Z score of each gene was then compared among all samples to determine which sample were highly expressed (Z score > 1.96).

**Orthology inference and molecular phylogenetic analyses.** Gene models of *A. digitifera*[32], *A. millepora*[95], *M. cactus*[32], *M. efflorescens*[32], *Astreopora myriophthalma*[32], *P. australiensis*[96], *S. pistillata*[97], *Orbicella faveolata*[98], *Pocillopora damicornis*[99], *E. diaphana*[100], *Nematostella vectensis*[101], *Drosophila melanogaster*[102], and *Homo sapiens*[103] were used for phylogenetic analyses in addition to those of *A. tenuis*. For *A. millepora*, *S. pistillata*, *O. faveolata*, *P. damicornis*, *N. vectensis*, *D. melanogaster*, and *H. sapiens*, we downloaded gene models from NCBI RefSeq (GCF_004143615.1 for *A. millepora*, GCF_002571385.1 for *S. pistillata*, GCF_002042975.1 for *O. faveolata*, GCF_003704095.1 for *P. damicornis*, GCF_001417965.1 for *E. diaphana*, GCF_000209225.1 for *N. vectensis*, GCF_000001215.4 for *D. melanogaster*, and GCF_000001405.39 for *H. sapiens*). For two *Montipora* and *Astreopora*, curated versions of gene models[94] were used. The longest transcript variants from each gene were selected and translated into protein sequences with TransDecoder v5.5.0 (https://github.com/TransDecoder/TransDecoder/wiki). Orthologous amino acid sequences (orthogroups) were identified using Orthofinder v2.5.4[104] with default settings, as described in Yoshioka et al.[94]. Orthologous amino acid sequences were aligned using MAFFT v7.310[105] with the "E-INS-i" strategy and gaps in alignments were removed using tri-mAL v1.2[106] with the "gappyout" option. After removing gaps, maximum likelihood analyses were performed using RAxML v8.2.10[107] with "bootstrap 100" and "protgammaauto" options.

**Reporting summary.** Further information on research design is available in the Nature Portfolio Reporting Summary linked to this article.

## Data availability

Raw RNA sequencing data for *A. tenuis* primary polyps inoculated with *S. microadriaticum* and with no exposure of algal cells are deposited in the DDBJ/EMBL/GenBank databases under accession number DRA014957 (BioProject accession: PRJDB8332).

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

## Acknowledgements

This study was supported by JSPS KAKENHI grants (20H03235 and 20K21860 for CS, 18H02270 and 21H04742 for HY, and 20H03066 for GS) and Grant-in-Aid for JSPS Fellows to YY (20J21301 and 23KJ2129). Computations were partially performed on the NIG supercomputer at ROIS National Institute of Genetics.

## Author contributions

C.S. conceived the project. G.S. and C.S. performed coral sampling. H.Y. maintained *S. microadriaticum* culture strain. C.Y.L. performed inoculation experiments. Y.Y. performed molecular biological experiments with help by T.U. Y.Y. performed bioinformatic analyses and wrote the main manuscript. C.S. supervised the project and edited the main manuscript. All authors checked and commented on the manuscript.

## Competing interests

The authors declare no competing interests.
