## [Peer Review File · Communications Biology]

Reviewers' comments:

Reviewer #1 (Remarks to the Author):

Yoshioka et al. conducted a study to explore the transcriptomic responses of *Acropora tenuis* during early life stages following inoculation with the native algal symbiont. By comparing the differentially expressed genes with the algal number, they also identified a set of potential "core" symbiosis-related genes. Overall, I find the research topic intriguing and the methodology sound. However, there are several significant points that need to be addressed before the manuscript can be considered for publication.

Major points:

1. I have some concerns about defining core symbiosis genes solely based on the correlation between the algal numbers and gene expression. It is important to consider that certain genes may need to be repressed for symbiosis formation, resulting in a lack of positive correlation with algal numbers. Additionally, the sample size of $n=5$ seems insufficient for conducting a reliable Pearson correlation analysis. For instance, when examining CSRP3 in Fig2. C, we observe minimal expression before 10dpi, followed by a sudden significant increase at 20 dpi, yet it still exhibits a high correlation ($r=0.973$) with the algal number.
2. Line 178-Line 192, can you make it clearer why some genes are considered as cluster with homologs of humans while others are not.
3. In the phylogenetic analysis, please be aware that *Nematostella* is a non-symbiotic cnidarian. The core genes don't necessarily to be conserved in *Nematostella*.
4. Line 261-Line 262, "the number of algae in host corals continues to increase in early life stages (Table 1), indicating that algal growth is more active" It is not necessary to be true. It could simply be due to the accumulation of engulfed algae from the environment.
5. Table S2, gene id alone didn't provide a lot of information for the field. It is better to include the log₂ fold change and domain information.
6. I am surprised by the relatively low number of differentially expressed genes (DEGs), which amounts to only 9 and 58, respectively when Smic is inoculated at the primary polyp stage. This discrepancy is quite notable compared to the inoculation at larval stages, where a much larger number of DEGs were observed. It would be valuable to include a thorough discussion addressing this disparity and exploring potential reasons for the observed difference in gene expression response between the two stages.

Minor points:

1. Fig. 1, please report how many replicates.
2. Fig. 2b please provide the alga number for each condition
3. Fig. 3, It's hard to read the domain information with so many color codes. It is better to directly label the domain names in the figure.
4. The abbreviation of species name in Fig3.b should be explained in the figure legend. Please also explain how the species tree is constructed.

Reviewer #2 (Remarks to the Author):

In this study, Yoshioka and co-authors investigated the gene expression changes in response to symbiosis at the early stage of *Acropora tenuis* primary polyps. Using RNA-seq, the authors identified a group of so-called core symbiosis-associated genes. These genes were then individually interpreted based on their phylogenetic association with genes from other model organisms. In a few cases, some of the genes were found duplicated at their genomic loci. With the phylogenetic analysis, the authors claim that some of these "core" symbiosis-related genes originated from gene duplication in the *Acropora* lineage.

The technique used in this study is RNA-seq and associated basic data analysis. Some phylogenetic analyses were briefly touched on at the end of the manuscript. Overall, the study provided some useful insights into potential key genes involved in the symbiosis establishment at the early life stage of corals. However, the manuscript would need some additional works to further improve the conciseness and clarity.

1. The authors claimed that a group of "core" genes were identified in this study. However, the wording of "core" is not clear. Usually, when talking about potential core genes associated with a biological process, it requires cross-study integration or meta-analysis. I would suggest the authors tone down and describe exactly what the study is looking into.
2. The authors need to improve the method section and give more details. For example, from the current context, it is hard to tell how many replicates were used in the RNA-seq experiments.
3. The study was explained as it focuses on algal symbiosis in primary polyps of *A. tenuis*. However, the authors suddenly introduced extra data from another study. Without proper rationale, this breaks the flow and makes it messy and hard to understand. If the authors wanted to focus on the primary polyp stage, I think it makes more sense to just focus on the genes identified from that stage to avoid confusion.
4. The authors claimed that the identified DEGs are core symbiosis-related genes. But from Fig. 2, it is easy to argue that most of these genes are actually more associated with certain development stages.
5. The authors should also spend more effort on interpreting their results. Although the paper identifies DEGs and discusses their differential expression, it lacks a comprehensive interpretation of the functional implications of these genes in the context of symbiotic interactions. Providing a more detailed discussion of the potential roles of these genes and their involvement in key physiological processes related to symbiosis would enhance the understanding of the findings. Additionally, functional enrichment analysis could be performed to identify overrepresented gene ontology terms or pathways among the DEGs, providing further insights into the biological processes affected by symbiosis.
6. Many of the identified DEGs were reported to be associated with symbiont number in primary polyps. However, there is no detailed method for how the correlation analysis was done. Hence, it is hard to exclude the possibility that some of the genes might be correlated with development stages since the symbiont numbers per polyp are entangled with the polyp stages.
7. I would also suggest that the author to further polish the writing, which would definitely be helpful in improving the overall quality of the manuscript.

for example:

Upregulation of sugar transporters under illuminated conditions in *Acropora* colonies 37 and in alga-containing cells of the coral *S. pistillata* 36 have been reported, suggesting that sugars are taken up by host corals via sugar transporters are an important nutrient source for *Acropora* in early life stages.

Could be rephrased to increase clarity:

Upregulation of sugar transporters under illuminated conditions has been reported in *Acropora* colonies 37 and in alga-containing cells of the coral *S. pistillata* 36, suggesting that sugars, taken up

by host corals via sugar transporters, are an important nutrient source for *Acropora* in early life stages.

Reviewer #1 (Remarks to the Author):

Yoshioka et al. conducted a study to explore the transcriptomic responses of *Acropora tenuis* during early life stages following inoculation with the native algal symbiont. By comparing the differential expressed genes with the alga number, they also identified a set of potential "core" symbiosis-related genes. Overall, I find the research topic intriguing and the methodology sound. However, there are several significant points that need to be addressed before the manuscript can be considered for publication.

We appreciate the positive assessments of our work and fruitful comments that greatly improved our manuscript. We have carefully revised the manuscript, as suggested by the reviewer. Please see our point-by-point responses below.

Major points:

1. I have some concerns about defining core symbiosis genes solely based on the correlation between the algal numbers and gene expression. It is important to consider that certain genes may need to be repressed for symbiosis formation, resulting in a lack of positive correlation with algal numbers. Additionally, the sample size of $n=5$ seems insufficient for conducting a reliable Pearson correlation analysis. For instance, when examining CSRP3 in Fig2. C, we observe minimal expression before 10dpi, followed by a sudden significant increase at 20 dpi, yet it still exhibits a high correlation ($r=0.973$) with the algal number.

First, we understand the reviewer's concern about definition of "core" symbiosis genes. Not only correlation analysis, but also extensive cross-study integration or meta-analysis is required to define "core" genes. Accordingly, we removed the term "core" and changed the title to "Genes possibly related to symbiosis in early life stages of *Acropora tenuis* inoculated with *Symbiodinium microadriaticum*."

We appreciate the comment that certain genes may need to be repressed for symbiosis formation. We found that no genes shared with planula larvae and primary polyps inoculated with Smic were negatively correlated with algal number in this study. To clarify this, we have added results of correlation analysis of the 17 genes in Supplementary Figure 1 and we described them (in red letters) in the current manuscript (L 120–122): "Expression levels of 15 DEGs showed a positive correlation with numbers of algae in corals ($r > 0.6$), while the remaining two genes showed no correlation (Figure 2c; Supplementary Figure 1)."

To compensate for the small sample size for correlation analysis, we performed validation of symbiosis-related genes by comparing results with two developmental stages, implying that expressed genes shared by planula larvae and primary polyps inoculated with Smic are more reliable, even with the small sample size for correlation analysis, as described in lines 116–120: "When we compared DEG repertoires between planula larvae and primary polyps inoculated with Smic, 17 DEGs were

common between them (Figure 2a). Furthermore, to test whether expression levels of these DEGs are correlated with numbers of algal symbionts, we performed correlation analysis between numbers of algal cells in corals (Figure 2b; Supplementary Table 1) and their expression levels.” However, as suggested by the reviewer, the results of correlation analyses alone may not be sufficient to claim that the genes specifically detected in primary polyps are symbiosis-related genes. Due to difficulties in answering this concern, we have removed sentences regarding correlation analysis of genes specifically detected in primary polyps to tone down the text in lines 132–152.

2. Line 178-Line 192, can you make it clearer why some genes are considered as cluster with homologs of humans while others are not.

Our explanation may have caused misunderstanding. We searched for potential homologs in humans or other model organisms for phylogenetic analysis to verify their evolutionary relationships. To clarify this, we have changed the sentences (in red letters) to “We next analyzed whether their potential homologs are present in model organisms (human, fruit flies, or yeast). Among the 12 genes, 10 had homology to human genes.” in lines 157–159. We hope this change makes it clear.

3. In the phylogenetic analysis, please be aware that *Nematostella* is a non-symbiotic cnidarian. The core genes don't necessarily to be conserved in *Nematostella*.

As noted by the reviewer, some of these genes may have developed functions involved in symbiosis after the divergence of corals and sea anemone (500 Mya) or they may have been specifically acquired in the coral lineage (coral restricted genes). Thus, whether the genes are conserved in *Nematostella* or not was not considered in this study, and we performed phylogenetic analysis to reveal their evolutionary history. It is coincidental that phylogenetic analyses in Supplementary Figures included *Nematostella* genes (All possible symbiosis-related coral genes, excluding *aten_s0218.g19* and *aten_s0219.g19*, have *Nematostella* homologs).

4. Line 261-Line 262, “the number of algae in host corals continues to increase in early life stages (Table 1), indicating that algal growth is more active” It is not necessary to be true. It could simply be due to the accumulation of engulfed algae from the environment.

We think that our original sentence is more likely because the number of Smic algae per coral larvae/polyp continued to increase even after the addition of algae was stopped (lines 89–93) and in our previous study using planula larvae. However, in order to accommodate the possibility that the reviewer indicated, we modified the sentences (in red letters) to “due to algal proliferation, the number of algae in host corals continues to increase until it reaches a certain number in early life stages (Figure 2b; Supplementary Table 1) or the accumulation of engulfed algae from the environment, indicating that the total amount of ROS also increases” in lines 288–291.

5. Table S2, gene id alone didn't provide a lot of information for the field. It is better to include the log2 fold change and domain information.

We agree and appreciate the comment. We have now added the information in Supplementary Data 1.

6. I am surprised by the relatively low number of differentially expressed genes (DEGs), which amounts to only 9 and 58, respectively when Smic is inoculated at the primary polyp stage. This discrepancy is quite notable compared to the inoculation at larval stages, where a much larger number of DEGs were observed. It would be valuable to include a thorough discussion addressing this disparity and exploring potential reasons for the observed difference in gene expression response between the two stages.

We appreciate that the reviewer properly recognized the important point in this study. According to this comment, we have added a new section, entitled, "Validation of symbiosis-related genes with two developmental stages inoculated with Smic" in the discussion section (L 204–222) as follows: "In this study, we identified 9 and 58 DEGs in primary polyps inoculated with Smic (Figure 1b). In contrast to the low numbers of DEGs, a larger number of DEGs were identified in planula larvae inoculated with Smic (Supplementary Data 1). A possible explanation for this is the presence of large number of genes whose expression levels are controlled by developmental stage or age (days), not by algal infection (Supplementary Figure 3). On the other hand, Yuyama et al.³⁰ reported differential expression of 7,667 genes in *A. tenuis* primary polyps inoculated with *D. trenchii*, a much higher number of DEGs than in this study. Although it is difficult to directly compare studies using different algal symbiont types, it is important to note that algal infection affects circadian rhythms of coral hosts regardless of algal species³⁴. In addition, algal culture media contain different types of nutrients and prolonged exposure to residual algal culture media may cause significant differences in polyp development. Indeed, proliferation by budding was observed in primary polyps inoculated with *D. trenchii*, whereas no budding occurred in apo-symbiotic polyps³⁰. Note that algal culture media was replaced with ASW during inoculation experiments in this study (see Methods). Comparing gene expression between samples in different states poses a challenge in determining whether observed changes in gene expression are truly associated with symbiosis. Therefore, it is crucial to validate results using other developmental stages to identify genes involved in coral symbiosis, and the 15 genes that commonly upregulated in planula larvae and primary polyps inoculated with Smic are likely associated with symbiosis in early life stages of *A. tenuis*."

Minor points:

1. Fig. 1, please report how many replicates.

We have modified Figure 1 and added a brief explanation to the legend.

2. Fig. 2b please provide the alga number for each condition

We have added algal number in corals in Figure 2, and raw data are shown in Supplementary Table 1.

3. Fig. 3, It's hard to read the domain information with so many color codes. It is better to directly label the domain names in the figure.

We now revised Figure 3. In addition, Supplementary Figure 3 was also directly labeled.

4. The abbreviation of species name in Fig3.b should be explained in the figure legend. Please also explain how the species tree is constructed.

As suggested by the reviewer, we wrote the information in the legend of Figure 3.

Reviewer #2 (Remarks to the Author):

In this study, Yoshioka and co-authors investigated the gene expression changes in response to symbiosis at the early stage of *Acropora tenuis* primary polyps. Using RNA-seq, the authors identified a group of so-called core symbiosis-associated genes. These genes were then individually interpreted based on their phylogenetic association with genes from other model organisms. In a few cases, some of the genes were found duplicated at their genomic loci. With the phylogenetic analysis, the authors claim that some of these “core” symbiosis-related genes originated from gene duplication in the *Acropora* lineage. The technique used in this study is RNA-seq and associated basic data analysis. Some phylogenetic analyses were briefly touched on at the end of the manuscript. Overall, the study provided some useful insights into potential key genes involved in the symbiosis establishment at the early life stage of corals. However, the manuscript would need some additional works to further improve the conciseness and clarity.

We appreciate the reviewer’s understanding of the value of our manuscript. We also appreciate the comments, which improved our manuscript. We have incorporated the suggestions to the extent possible. Please see our point-by-point responses below.

1. The authors claimed that a group of “core” genes were identified in this study. However, the wording of “core” is not clear. Usually, when talking about potential core genes associated with a biological process, it requires cross-study integration or meta-analysis. I would suggest the authors tone down and describe exactly what the study is looking into.

We agree with the reviewer that cross-study integration or meta-analysis is required to use the term “core.” Thus, we have removed “core” and changed the title to, “Genes possibly related to symbiosis in early life stages of *Acropora tenuis* inoculated with *Symbiodinium microadriaticum*” in the revised manuscript. In addition, we describe the study objective.

2. The authors need to improve the method section and give more details. For example, from the current context, it is hard to tell how many replicates were used in the RNA-seq experiments.

We apologize for the lack of clarity in Methods. We have overhauled all sentences in the Methods and have provided more detail (in red letters) as follows.

Lines 343–345: “Two days after metamorphosis, we divided primary polyps into two treatment groups, with at least two wells per treatment. One well containing ~5 individuals was considered one replicate in this study (Figure 1a).”

Lines 348–350: “2 mL of algae-containing ASW (200,000 algal cells) were added to the first group of primary polyps (treatment group). For the remaining group, 2 mL of ASW were added, which was used as a control group (apo-symbiotic).”

Lines 359–360: “At 10 and 20 dpi, ~5 primary polyps from each well of 24-well dishes were

collected (n = 3 for 10 dpi and n = 2 for 20 dpi)”

Lines 371–372: “Transcript abundances in each sample were quantified using SALMON v1.5.2⁸⁸ with default settings.”

Lines 386–389: “Data processing was performed as in Yoshioka et al.³⁴. Briefly, RNA-seq reads were mapped to *A. tenuis* gene models after trimming of low-quality reads and Illumina sequence adaptors. Gene expression levels of Smic-inoculated samples were compared to their corresponding apo-symbiotic samples to identify DEGs.”

Lines 390–392: “These DEG repertoires were compared with those identified in *A. tenuis* primary polyps inoculated with Smic. A list of DEGs is provided in Supplementary Data 1.”

Lines 398–400: “Relative gene expression levels of symbiotic planula larvae and primary polyps are provided in Supplementary Data 1. Numbers of algal cells in corals are provided in Supplementary Table 1.”

Lines 409–410: “Data processing was performed as in Yoshioka et al.⁹⁴. Briefly, low-quality reads and sequencing adaptors were trimmed, and cleaned reads were mapped to *A. tenuis* gene models.”

Lines 411–413: “Gene expression levels (transcripts per million, TPM) were estimated with SALMON and were transformed into Z-scores. The Z-score of each gene was then compared among all samples to determine in which sample were highly expressed (Z-score > 1.96).”

s

3. The study was explained as it focuses on algal symbiosis in primary polyps of *A. tenuis*. However, the authors suddenly introduced extra data from another study. Without proper rationale, this breaks the flow and makes it messy and hard to understand. If the authors wanted to focus on the primary polyp stage, I think it makes more sense to just focus on the genes identified from that stage to avoid confusion.

We appreciate the helpful comments to improve the flow of the text. In this study, we attempted to identify possible symbiosis-related genes in *Acropora* early life stages by comparing both planula and polyp stages. Thus, we revised the Introduction (L 73–82): “In addition, it is challenging to distinguish between genes associated with symbiosis and genes involved in development in corals that acquire algal symbionts horizontally in early life stages. It is expected that key symbiosis-related genes would be continuously expressed or suppressed when corals acquire algal symbionts. Therefore, it is crucial to validate the candidate genes involved in symbiosis with multiple developmental stages. In the present study, we performed inoculation experiments with Smic using *A. tenuis* primary polyps and analyzed transcriptomes of *A. tenuis* primary polyps inoculated with Smic to reveal transcriptomic responses of primary polyps to Smic inoculation. Then we compared transcriptomic responses of *A. tenuis* planula larvae and primary polyps inoculated with Smic to identify genes associated with symbiosis in *A. tenuis* early life stages.”, and Result section (L 110–111): “In order to identify genes

associated with symbiosis shared by multiple developmental stages of *A. tenuis*, we compared repertoires of genes whose expression levels were significantly...”

4. The authors claimed that the identified DEGs are core symbiosis-related genes. But from Fig. 2, it is easy to argue that most of these genes are actually more associated with certain development stages.

We understand the reviewer’s concern that these genes may be related to development, rather than symbiosis. In order to remove genes specifically expressed at certain developmental stages, we focused on genes altered by *Smic* inoculation that were shared by planula larvae and primary polyp stages in the revised Figure 2.

5. The authors should also spend more effort on interpreting their results. Although the paper identifies DEGs and discusses their differential expression, it lacks a comprehensive interpretation of the functional implications of these genes in the context of symbiotic interactions. Providing a more detailed discussion of the potential roles of these genes and their involvement in key physiological processes related to symbiosis would enhance the understanding of the findings. Additionally, functional enrichment analysis could be performed to identify overrepresented gene ontology terms or pathways among the DEGs, providing further insights into the biological processes affected by symbiosis.

We agree that this manuscript lacks a comprehensive interpretation of functional implications in the context of symbiotic interactions in the Discussion. In the section “Metabolic interactions during symbiosis in early *Acropora tenuis* life stages”, we added introductory sentences in lines 240–243 and 259–261, and we modified sentences in lines 247–249 to highlight possible functions of these genes. In the section “Possible oxidative stress prevention and immune control during symbiosis”, we have extensively revised sentences in lines 293–310 to discuss the importance of the immune system and possible functions of identified genes. In addition, we summarized functional implications of possible symbiosis-related genes in Figure 4.

For additional functional enrichment analysis, as mentioned above, we have focused on DEGs shared by planula larvae and primary polyps inoculated with *Smic*, resulting 15 possible symbiosis-related genes. Basically, identification of overrepresented gene ontology terms or pathways is suitable for analysis of hundreds or thousands of DEGs. We identified 15 genes, which is too small a number for statistical analysis in enrichment analysis, but it is possible to carefully annotate each gene as performed in this study. Therefore, we believe that identification of overrepresented gene ontology terms or pathways for the 15 genes is not necessary.

6. Many of the identified DEGs were reported to be associated with symbiont number in primary polyps. However, there is no detailed method for how the correlation analysis was done. Hence, it is

hard to exclude the possibility that some of the genes might be correlated with development stages since the symbiont numbers per polyp are entangled with the polyp stages.

We apologize for the lack of detailed methods for the correlation analysis. As noted by the reviewer, it is difficult to exclude the possibility that some genes specifically detected in primary polyps might be associated with developmental stage. Thus, we removed sentences regarding correlation analysis of genes specifically detected in primary polyps.

For correlation analysis of DEGs common to planula larvae and primary polyps inoculated with *Smic*, we have included a plot showing the correlation in Supplementary Figure 1, a table showing the exact number of acquired *Smic* in Supplementary Table 1, and a table showing the relative expression levels in Supplementary Data 1. These were mentioned in the current manuscript in addition to the description in the method section “Correlation analysis between gene expression levels and algal densities in corals” (L 394–400): “Relative gene expression levels of symbiotic planula larvae and primary polyps compared to their corresponding apo-symbiotic larvae and polyps were calculated using EdgeR. Correlation between \log_2 (relative gene expression levels of corals inoculated with *Smic*) and \log_2 (algal cell numbers in corals) were examined using the function “cor.test” in R⁹⁰. Genes with correlation coefficient > 0.6 were considered positively correlated with algal number in corals. Relative gene expression levels of symbiotic planula larvae and primary polyps are provided in Supplementary Data 1. Numbers of algal cells in corals are provided in Supplementary Table 1.”

7. I would also suggest that the author to further polish the writing, which would definitely be helpful in improving the overall quality of the manuscript.

for example:

Upregulation of sugar transporters under illuminated conditions in *Acropora* colonies 37 and in alga-containing cells of the coral *S. pistillata* 36 have been reported, suggesting that sugars are taken up by host corals via sugar transporters are an important nutrient source for *Acropora* in early life stages.

Could be rephrased to increase clarity:

Upregulation of sugar transporters under illuminated conditions has been reported in *Acropora* colonies 37 and in alga-containing cells of the coral *S. pistillata* 36, suggesting that sugars, taken up by host corals via sugar transporters, are an important nutrient source for *Acropora* in early life stages.

As suggested, we tried to improve our writing as much as possible, and the manuscript has been re-reviewed by a technical editor.

REVIEWERS' COMMENTS:

Reviewer #1 (Remarks to the Author):

The revised manuscript's flow has noticeably improved. The change from "symbiosis core gene" to "symbiosis-related genes" makes the conclusion more reasonable. I find myself largely satisfied with the responses to most of my comments. However, there are two minor points I'd like to address, which could potentially contribute to further enhancing the manuscript:

In the discussion about the DEG numbers under different experimental settings, while the author's provided explanations are valid, it's worth considering an additional perspective. It's possible that a substantial portion of the DEGs in other settings might arise as indirect effects of the algae inoculation. In contrast, the relatively low number of DEGs observed in the primary polyp stage could indeed represent genes directly linked to the algae inoculation. This notion corresponds well with the observation that several of these genes exhibit high expression in alga-containing cells, as demonstrated in the single-cell RNA-seq analysis of *Stylophora pistillata* and *Xenia*.

Line 215: The sentence "proliferation by budding was observed in primary polyps" could be refined by replacing "proliferation" with "asexual reproduction." This would align the terminology more accurately with the context.

Reviewer #2 (Remarks to the Author):

Thanks for making all the changes in response to my and another reviewer's comments. With these changes, the clarity and conciseness of the manuscript have been significantly improved. And I agree with the authors that further enrichment analysis won't provide much information with such a small list of DEGs. The identified DEGs could serve as a good reference list for future functional investigations looking into coral-algae symbiosis, especially at the early life stage of coral hosts and at their initial interactions between host and symbiont.

One quick comment: there seems some formatting issues, not sure if it is from the authors' side or from the submission system.

For example:

- 1) line 379, the writing of $1e-5$. I believe "-5" should be normal text, not to be superscript, unless the authors want to write it as 10^{-5} .
- 2) lines 417 and 418, all the references cited in these two lines appeared twice on my side. Please double check.

Reviewer #1 (Remarks to the Author):

The revised manuscript's flow has noticeably improved. The change from "symbiosis core gene" to "symbiosis-related genes" makes the conclusion more reasonable. I find myself largely satisfied with the responses to most of my comments. However, there are two minor points I'd like to address, which could potentially contribute to further enhancing the manuscript. In the discussion about the DEG numbers under different experimental settings, while the author's provided explanations are valid, it's worth considering an additional perspective. It's possible that a substantial portion of the DEGs in other settings might arise as indirect effects of the algae inoculation. In contrast, the relatively low number of DEGs observed in the primary polyp stage could indeed represent genes directly linked to the algae inoculation. This notion corresponds well with the observation that several of these genes exhibit high expression in alga-containing cells, as demonstrated in the single-cell RNA-seq analysis of *Stylophora pistillata* and *Xenia*.

We greatly appreciate the reviewer's suggestion. We have added the following sentences to discussion in lines 208–210: “**In other words, a substantial portion of DEGs identified in planula larvae may have arisen as an indirect effect of algal inoculation. In contrast, DEGs identified in primary polyps may represent genes directly related to algal inoculation.**”

Line 215: The sentence "proliferation by budding was observed in primary polyps" could be refined by replacing "proliferation" with "asexual reproduction." This would align the terminology more accurately with the context.

We agree and we revised the sentence to, “**asexual reproduction by budding**” (Line 219)

Reviewer #2 (Remarks to the Author):

Thanks for making all the changes in response to my and another reviewer's comments. With these changes, the clarity and conciseness of the manuscript have been significantly improved. And I agree with the authors that further enrichment analysis won't provide much information with such a small list of DEGs. The identified DEGs could serve as a good reference list for future functional investigations looking into coral-algae symbiosis, especially at the early life stage of coral hosts and at their initial interactions between host and symbiont. One quick comment: there seems some formatting issues, not sure if it is from the authors' side or from the submission system.

For example:

- 1) line 379, the writing of $1e-5$. I believe "-5" should be normal text, not to be superscript, unless the authors want to write it as 10^{-5} .
- 2) lines 417 and 418, all the references cited in these two lines appeared twice on my side. Please double check.

We are grateful for the positive assessments of our work and greatly appreciate the careful reading of the manuscript. We revised all of them in the current manuscript.